# Frequent Usage of Convenience Stores is Associated with Low Diet Quality

**DOI:** 10.3390/nu11061212

**Published:** 2019-05-28

**Authors:** Ayumi Kaji, Yoshitaka Hashimoto, Ryosuke Sakai, Hiroshi Okada, Masahide Hamaguchi, Emi Ushigome, Saori Majima, Masahiro Yamazaki, Michiaki Fukui

**Affiliations:** 1Department of Endocrinology and Metabolism, Graduate School of Medical Science, Kyoto Prefectural University of Medicine, Kyoto 602-8566, Japan; kaji-a@koto.kpu-m.ac.jp (A.K.); sakaryo@koto.kpu-m.ac.jp (R.S.); conti@koto.kpu-m.ac.jp (H.O.); mhama@koto.kpu-m.ac.jp (M.H.); emis@koto.kpu-m.ac.jp (E.U.); saori-m@koto.kpu-m.ac.jp (S.M.); masahiro@koto.kpu-m.ac.jp (M.Y.); michiaki@koto.kpu-m.ac.jp (M.F.); 2Department of Diabetes and Endocrinology, Matsushita Memorial Hospital, Moriguchi, Osaka 570-8540, Japan

**Keywords:** convenience store, diet quality, food environment, habitual diet intake, lifestyle, type 2 diabetes

## Abstract

Previous studies have revealed that the density of convenience stores in the neighborhood was associated with chronic diseases. In Japan, convenience stores are more common, and it is thus more important to assess whether people use convenience stores than the density or availability of the convenience stores. In this cross-sectional study of patients with type 2 diabetes, the association between the usage of the convenience stores and dietary habits or the prevalence of hypertension was evaluated. Among the 206 men and 161 women in the study, 24 men and 9 women used convenience stores three or more times per week. Fruit and vegetable intake (men, 132 (102−191) vs. 192 (128−267) g/1000 kcal, *p* = 0.019; and women, 178 (132−207) vs. 239 (172−313) g/1000 kcal, *p* = 0.063) of patients who frequently use convenience stores was lower compared to those who did not. Net endogenous acid production score (men, 55.2 (45.4−65.2) vs. 48.9 (42.3−56.8) mEq/day, *p* = 0.013; and women, 56.9 (52.6−59.8) vs. 46.3 (40.9−54.0) mEq/day, *p* = 0.050) and intake of carbohydrate to fiber ratio (men, 21.5 (20.0−29.3) vs. 19.9 (15.7−25.0), *p* = 0.052; and women, 21.0 (18.9−23.9) vs. 16.2 (13.8−20.3), *p* = 0.017) of patients who frequently use convenience stores were higher compared to those who did not. Additionally, frequent usage of convenience stores was associated with the prevalence of hypertension after adjusting for covariates (5.01; 95% confidence interval, 1.12−22.50; *p* = 0.035). In conclusion, frequent usage of convenience stores is associated with low diet quality and the prevalence of hypertension.

## 1. Introduction

Currently, lifestyles have become diverse and more irregular, and food consumption patterns are changing because of this lifestyle change [1,2]. One of the factors in this change is neighborhood food environment. In fact, having a variety of neighborhood food outlets was associated with beneficial dietary patterns [3], and that the increased number of fast-food outlets in the neighborhood was associated with increased risk of type 2 diabetes and obesity [4]. Furthermore, it has been reported that the high density of convenience store in the neighborhood was associated with obesity [5]. In this way, neighborhood food environment is associated with chronic diseases.

On the other hand, a healthy diet is important to prevent various lifestyle diseases. For example, the Dietary Approaches to Stop Hypertension (DASH) diet reduces blood pressure, and it is recommended in preventing the occurrence of cardiovascular diseases [6]. In a previous study, we showed that the low carbohydrate to fiber ratio in diet was associated with the prevalence of metabolic syndrome in patients with type 2 diabetes [7]. Moreover, higher dietary fiber intake was associated with lower risk of developing diabetes, obesity, hypertension, and coronary artery disease [8].

Several of previous studies on convenience stores, which are one of the components of the neighborhood food environment, mainly investigated density or the availability. Unlike in other countries, convenience stores are more common in Japan and people living in Japan can easily access them. In the United States, for example, the habitable land area that was land area minus forest area was 610,720,000 ha (State of the World’s Forests, http://www.fao.org/3/i2000e/i2000e.pdf), and the convenience store count was reported to be 154,958 stores (https://www.nacsmagazine.com/issues/february-2018/us-convenience-stores-continue-growth); hence, there are about 0.25 convenience stores per 1000 ha. On the contrary, in Japan, the habitable land area and the convenience store count were 11,471,000 ha and 56,374 stores, respectively. Thus, there are about 4.91 convenience stores per 1000 ha. In summary, Japan has about twenty times more convenience stores than the United States. Therefore, to investigate the impact of food consumption patterns on dietary habits and peoples’ health, it is necessary to check, not the density or the availability, but the frequency of convenience store usage, especially in Japan. In addition, diet is an essential treatment target for patients with type 2 diabetes [9]. Thus, to investigating the convenience store usage in patients with type 2 diabetes is important. However, the association between the frequency of convenience store usage and dietary habits, especially in type 2 diabetes, remains to be elucidated. Thus, we investigated this association in this cross-sectional study.

## 2. Materials and Methods

### 2.1. Patients and Study Design

This was a cross-sectional study as part of the KAMOGAWA-DM cohort study that has continued since 2014 in order to confirm the natural history of diabetes [10,11,12]. In this study, we included outpatients of the Department of Endocrinology and Metabolism at Kyoto Prefectural University of Medicine Hospital and the Department of Diabetology at Kameoka Municipal Hospital who answered questionnaires from January 2016 to May 2018.

Among the 438 patients (244 men and 194 women) who answered both the brief-type self-administered diet history questionnaire (BDHQ) and the questionnaire about convenience stores, we excluded the following 29 patients with the following characteristics: incomplete questionnaires (14 men and 7 women), extremely low or high energy intake (<600 or >4000 kcal/day) [13] (3 men and 4 women) because these calorie intakes are unreliable, no data in biochemical tests (one man), and no type 2 diabetes (20 men and 22 women). Thus, the final study population was 367 patients (206 men and 161 women).

The study was approved by the local research ethics committee (No. RBMR-E-466-5) and was conducted in accordance with the Declaration of Helsinki. All of the patients signed the written informed consent form before inclusion.

### 2.2. Questionnaire About Dietary Habit

We assessed the patients’ habitual food and nutrient intake during the preceding 1-month period using a brief-type self-administered diet history questionnaire (BDHQ) [14,15]. The detail and validity of BDHQ were reported previously [16,17].

Using the estimated intake by BDHQ, we defined “carbohydrate to fiber ratio” as carbohydrate intake (g/day)/fiber intake (g/day), which was reported to be the marker for metabolic syndrome in our previous study [7]. We also estimated the potential renal acid load (PRAL) score and net endogenous acid production (NEAP) score as indicators of dietary acid load. PRAL and NEAP were calculated according to the following equations, respectively: PRAL (mEq/day) = 0.49 × protein (g/day) + 0.037 × phosphorus (mg/day) − 0.021 × potassium (mg/day) − 0.026 × magnesium (mg/day) − 0.013 × calcium (mg/day) [18] and NEAP (mEq/day) = (54.5 × protein (g/day) / potassium (mEq/day)) − 10.2 [19]. We defined greater than 20 g/day alcohol intake, which was estimated using BDHQ, as “habit of drinking alcohol.”

### 2.3. Questionnaire About Lifestyle Characteristics

This questionnaire contained questions about lifestyle characteristics, such as physical activity, smoking status, and the usage of convenience stores. “Habit of exercise” was defined as performing any kind of physical activity at least once a week. “Habit of smoking” was defined as currently smoking cigarettes or another tobacco product. Regarding the convenience store usage, we investigated whether each patient uses convenience store three or more times per week or not, since visiting fast-food restaurant three or more times per week caused significant changes in nutrient intake, and they cared about their health less compared to those who visit two or fewer times in the previous study [20]. We asked the following closed-ended question: “Did you buy meals from convenience stores three or more times per week?”

### 2.4. Patients’ Data

Blood samples and urine samples were collected in the morning for biochemical measurements, when the self-reports of dietary intake were performed. We considered the following laboratory parameters: hemoglobin A1c, plasma glucose, creatinine, glomerular filtration rate (GFR), uric acid, alanine aminotransferase, aspartate aminotransferase, total cholesterol, high-density lipoprotein cholesterol, triglycerides, and urinary albumin excretion. Hemoglobin A1c was assayed using high-performance liquid chromatography. GFR was estimated using the equation of the Japanese Society of Nephrology [21]: eGFR (ml/min/1.73 m^2^) = 194 × Cre^−1.094^ × age^−0.287^. For women, the eGFR was multiplied by a correction factor of 0.739. Urinary albumin excretion was measured using an immunoturbidimetric assay.

Blood pressure measurement was performed automatically using a HEM-906 device (OMRON, Kyoto, Japan) in a quiet space after 5 min of rest, when the self-reports of dietary intake were performed.

We obtained the data about medications from the electronic medical records. We checked whether each patient was on medications for diabetes, hypertension, and/or dyslipidemia.

### 2.5. Definition

According to the Classification and Diagnosis of Diabetes by the American Diabetes Association, we categorized patients into impaired glucose tolerance, type 1 diabetes, type 2 diabetes, and other specific types of diabetes [22]. Among them, patients with type 2 diabetes were included in our study population.

Hypertension was defined as systolic blood pressure ≥140 mmHg and/or diastolic blood pressure ≥90 mmHg and/or usage of medication for hypertension [23].

### 2.6. Statistical Analysis

We performed statistical analyses using EZR (version 1.36) [24], and *p* value < 0.05 was considered to be statistically significant. To investigate the distribution of variables, we performed a Shapiro-Wilk test. Continuous variables were described as the mean (standard deviation) or median (interquartile range), and categorical variables were presented as the number. The differences in characteristics and dietary intake between the patients with and without usage of convenience stores were evaluated using Student’s t-test, Mann–Whitney U test or chi-squared test. Since the characteristics and dietary intake were different between men and women, we separately analyzed the patients by gender. We performed logistic regression analysis to examine the effects of the convenience stores usage on the prevalence of hypertension. The following factors were considered as independent variables: age, sex, body mass index (BMI) [25], duration of diabetes, hemoglobin A1c [26], energy intake, salt intake [27], habit of drinking alcohol [28], habit of exercise [29], and habit of smoking [30].

## 3. Results

Study patients’ characteristics are summarized in Table 1. Median (interquartile range) age and BMI were 68.0 (62.5−74.0) years and 23.6 (21.7−26.3) kg/m^2^, respectively. Regarding men, 11.7% of the patients bought meals from convenience stores three or more times a week. Patients who use convenience stores frequently were significantly younger than patients who did not use convenience stores frequently (64.5 (54.0–67.3) vs. 69.0 (64.0–74.0) years, *p* = 0.004). The former were more likely to have a habit to smoke than the latter (37.5 vs. 18.1%, *p* = 0.052), and diastolic blood pressure (83 (76–90) vs. 75 (66–81) mmHg, *p* < 0.001) and systolic blood pressure (134 (127–148) vs. 130 (122–138) mmHg, *p* = 0.101) were also higher in the former than in the latter.

Regarding women, 5.6% patients bought meals from convenience stores frequently. Fasting plasma glucose values were lower in patients who frequently use convenience stores (6.6 (5.6–6.7) vs. 7.4 (6.2–8.6) mmol/L, *p* = 0.058) compared to those who do not frequently use convenience stores. Although patients who frequently use convenience stores were younger (65.0 (58.0–68.0) vs. 69.0 (62.0–74.0) years, *p* = 0.233), duration of diabetes tended to be longer (16.0 (9.0–20.0) vs. 12.0 (5.0–18.0) years, *p* = 0.355).

Table 2 shows the results of the dietary survey. Regarding men, patients who frequently use convenience stores consumed less protein compared with patients who did not frequently use convenience stores (14.7 (12.5–16.3) vs. 15.7 (13.7–17.8)%, *p* = 0.062), whereas energy intake was not significantly different (29.2 (21.6–38.1) vs. 29.4 (25.0–35.3) kcal/kg ideal body weight/day, *p* = 0.710). Fruit and vegetable intake (132 (102–191) vs. 192 (128–267) g/1000 kcal, *p* = 0.019), low-fat dairy product intake (0.0 (0.0–0.0) vs. 0.0 (0.0–58.0) g/1000 kcal, *p* = 0.009), and dietary pulse intake (23.6 (13.9–32.8) vs. 27.6 (17.2–43.5) g/1000 kcal, *p* = 0.093), which were adjusted for total energy intake by nutrient density method, were also less in the former than in the latter. Moreover, PRAL (12.8 (2.5–18.9) vs. 7.0 (0.1–14.2) mEq/day, *p* = 0.039) and NEAP (55.2 (45.4–65.2) vs. 48.9(42.3–56.8) mEq/day, *p* = 0.013) values of patients who frequently use convenience stores were significantly higher compared to those who did not frequently use convenience stores.

Regarding women, patients who had a habit to use convenience stores consumed significantly more carbohydrate (60.8 (51.5–64.5) vs. 51.1 (45.4–55.8)%, *p* = 0.048) and significantly less protein (15.0 (13.7–16.5) vs. 17.3 (15.7–20.2)%, *p* = 0.005), leading to a higher energy intake (39.3 (27.7–42.9) vs. 27.7 (22.8–33.5) kcal/kg ideal body weight/day, *p* = 0.071). In common with men, fruit and vegetable intake (178 (132–207) vs. 239 (172–313) g/1000 kcal, *p* = 0.063), low-fat dairy product intake (0.0 (0.0–0.0) vs. 0.0 (0.0–58.0) g/1000 kcal, *p* = 0.057), and dietary pulse intake (18.8 (10.5–31.4) vs. 38.8 (21.6–61.2) g/1000 kcal, *p* = 0.011) of women who frequently use convenience stores were less compared to those women who did not frequently use convenience stores. Simultaneously, intake of processed meats (8.3 (2.8–10.6) vs. 2.8 (1.6–7.2) g/1000 kcal, *p* = 0.140) and confectionery (41.9 (27.1–48.1) vs. 29.6 (13.9–45.4) g/1000 kcal, *p* = 0.154) were larger in patients who have a habit of using convenience stores. Moreover, intake of carbohydrate to fiber ratio (21.0 (18.9–23.9) vs. 16.2 (13.8–20.3), *p* = 0.017) of women who frequently use convenience stores was higher compared to that of women who did not frequently use convenience stores. 

The data presented in Table 3 show the results of logistic regression analysis on the prevalence of hypertension. In addition to age (odds ratio, 1.04; 95% confidence interval [CI], 1.01–1.07; *p* = 0.019) and BMI (1.19; 95% CI, 1.09–1.23; *p* < 0.001), usage of convenience stores three or more times per week (5.01; 95% CI, 1.12–22.50; *p* = 0.035) also correlated with the prevalence of hypertension.

## 4. Discussion

In this study, we researched the frequency of convenience store usage in patients with type 2 diabetes and verified the association between the frequent usage of convenience stores and habitual dietary intake. Our survey showed that 33 out of the 367 patients answered that they bought meals from convenience stores three or more times a week. Moreover, patients who use convenience stores frequently had a low quality diet, such as lower fruit and vegetable intake, low-fat dairy product intake, and dietary pulse intake, compared to those who did not frequently use convenience stores. These food items are recommended to be consumed in the DASH diet to prevent the onset of hypertension. In fact, usage of the convenience stores three or more times per week was associated with the prevalence of hypertension.

A study from the United States showed that the neighborhood availability of convenience stores was negatively associated with consumption of whole grains and diet quality score [31]. Furthermore, a study from Ireland showed that children who lived near the convenience store had lower dietary quality score compared to children who lived far from the convenience store [32]. The study from Australia also showed that children who have convenience stores or fast-food outlets in the neighborhood were less likely to consume fruits ≥2 times/day and vegetables ≥3 times/day [33]. In Japan, unlike in other countries including the United States, convenience stores are more common, and it is thus more important to assess whether people use convenience stores, rather than the density or availability of convenience stores. As expected, the dietary pattern was different between patients who frequently use convenience stores and patients who did not frequently use convenience stores, and the frequent usage might cause a lower quality of diet.

To consume fruits and vegetables properly has a protective effect on metabolic syndrome [34,35]. Among them, the risk of hypertension has been reported in many studies. The meta-analysis assessing nine cohort studies revealed an inverse dose–response relationship between fruit and vegetable intake and the risk of developing hypertension [36]. PRAL and NEAP are indicators of dietary acid load, and they have higher scores in diets which are rich in acidogenic foods such as meat and fish and deficient in alkaline foods such as fruits and vegetables [37]. In this study, patients who frequently bought meals from a convenience store had higher PRAL and NEAP scores compared to patients who did not frequently buy meals from a convenience store, and a higher score was reported to be associated with the risk of hypertension or total mortality [38,39].

The median age of the people who used convenience stores 3 or more times per week is 65, and this median age is younger than that of the people who did not use them. One possible reason why the median age of the people who use convenience stores 3 or more times per week is younger might be that the younger patients with type 2 diabetes find it difficult to cook for themselves. Moreover, we showed that the ratio of smokers in the patients who frequently use convenience stores was higher compared to the patients who did not frequently use convenience stores. Cigarette smoking was reported to cause a sharp increase in blood pressure in the short-term [40,41]. A large cohort study suggested that current smokers were at greatest risk of developing hypertension [42]. Thus, there is also the possibility that the higher ratio of hypertension in the patients who use convenience stores frequently is caused by smoking.

To the best of our knowledge, this is the first study of patients with type 2 diabetes that investigates the association between the usage of convenience stores and habitual dietary intake. We targeted patients with type 2 diabetes as their diet has important meaning, and habitual dietary intake was investigated in detail using a validated questionnaire. Moreover, the present study focused on how often patients really buy meals from convenience stores instead of the number or density of convenience stores, which were the focused of several previous studies.

However, our study also had some limitations. First, this is a cross-sectional study; thus, it is not possible for us to reveal a causal effect. Second, we did not investigate socioeconomic status of the patients, including income and education, which possibly influences convenience store usage. Third, we did not consider other food choices, such as fast-food, ready-made meals, take-out food, and eating out. Fourth, to compare the proportion of frequent usage of convenience stores in Japan with that of other countries is important. Unfortunately, however, no previous studies examined the frequent usage of convenience stores. In addition, the number of female convenience store users is small. Thus, results from only 9 individuals are likely not generalizable. Finally, this study included only Japanese patients; thus, the generalizability of our study to other ethnic populations is uncertain.

## 5. Conclusion

In conclusion, the frequent usage of convenience stores was associated with low diet quality and was associated with the prevalence of hypertension. Therefore, there is a possibility that to intervene in the usage of convenience stores might be one of the new important targets to prevent the development of hypertension by dietary treatment. 

## Figures and Tables

**Table 1 nutrients-11-01212-t001:** Clinical characteristics of study participants.

	Men			Women		
	Usage of Convenience Stores Three or More Times Per Week		*p*	Usage of Convenience Stores Three or More Times Per Week		*p*
*N*	(+) *n* = 24	(−) *n* = 182		(+) *n* = 9	(−) *n* = 152	
Age (years)	64.5 (54.0−67.3)	69.0 (64.0−74.0)	0.004	65.0 (58.0−68.0)	69.0 (62.0−74.0)	0.233
Body mass index (kg/m^2^)	23.3 (20.7−25.5)	23.6 (21.9−26.1)	0.399	23.4 (21.4−25.4)	23.8 (21.3−26.9)	0.716
Habit of exercise (−/+)	15/9	87/95	0.256	6/3	80/72	0.633
Habit of smoking (−/+)	15/9	149/33	0.052	7/2	143/9	0.229
Habit of drinking alcohol (−/+)	18/6	144/38	0.843	9/0	149/3	1.000
Hemoglobin A1c (mmol/mol)	51.9 (49.5−58.7)	53.0 (47.5−58.2)	0.658	54.1 (46.5−55.2)	51.9 (48.6−58.5)	0.627
Hemoglobin A1c (%)	6.9 (6.7−7.5)	7.0 (6.5−7.5)	0.658	7.1 (6.4−7.2)	6.9 (6.6−7.5)	0.627
Fasting plasma glucose (mmol/L)	7.4 (6.7−8.5)	7.7 (6.6−9.3)	0.816	6.6 (5.6−6.7)	7.4 (6.2−8.6)	0.058
Duration of diabetes (years)	11.0 (6.8−15.8)	14.0 (8.0−21.8)	0.249	16.0 (9.0−20.0)	12.0 (5.0−18.0)	0.355
Medication for diabetes (−/+)	2/22	16/166	1.000	0/9	11/141	0.876
Insulin treatment (−/+)	22/2	171/11	1.000	9/0	142/10	0.933
Creatinine (μmol/L)	69.8 (59.2−82.7)	77.8 (64.8−91.9)	0.198	51.3 (44.2−61.9)	56.6 (50.2−63.0)	0.466
eGFR (mL/min/1.73 m^2^)	80.2 (63.5−92.4)	68.7 (55.6−81.4)	0.077	78.5 (67.1−94.1)	71.2 (60.7−81.3)	0.326
UAE (mg/gCr)	23.7 (11.3−223.8)	19.6 (7.0−85.5)	0.391	13.8 (10.9−18.0)	18.6 (9.3−69.4)	0.299
Uric acid (μmol/L)	330.1 (282.5−376.2)	333.1 (279.6−386.6)	0.665	261.7 (178.4−267.7)	279.6 (237.9−333.1)	0.067
Alanine aminotransferase (IU/L)	20.5 (16.0−27.0)	19.0 (14.3−27.0)	0.875	16.0 (11.0−21.0)	15.5 (13.0−22.3)	0.886
Aspartate aminotransferase (IU/L)	21.0 (19.0−26.0)	21.0 (18.0−25.8)	0.711	19.0 (15.0−21.0)	19.0 (16.0−23.3)	0.443
Systolic blood pressure (mmHg)	134 (127−148)	130 (122−138)	0.101	134 (126−137)	131 (119−141)	0.760
Diastolic blood pressure (mmHg)	75.4 (9.8)	82.7 (12.8)	0.003	73.3 (10.5)	71.3 (12.2)	0.586
Medication for hypertension (−/+)	10/14	75/107	1.000	3/6	73/79	0.607
Total cholesterol (mmol/L)	4.6 (4.1−5.3)	4.7 (4.1−5.4)	0.831	5.2 (5.0−5.4)	5.2 (4.6−5.9)	0.979
HDL cholesterol (mmol/L)	1.42 (1.13−1.66)	1.37 (1.20−1.63)	0.961	1.58 (1.53−1.97)	1.54 (1.34−1.81)	0.399
Triglyceride (mmol/L)	1.42 (0.91−1.65)	1.30 (0.92−1.91)	0.965	1.25 (0.79−1.61)	1.29 (0.88−1.71)	0.635
Medication for dyslipidemia (−/+)	13/11	95/87	1.000	4/5	86/66	0.714

Data are expressed as mean (standard deviation), median (interquartile range) or number. eGFR, estimated glomerular filtration rate; UAE, urinary albumin excretion; HDL, high−density lipoprotein. “Habit of exercise” was defined as performing any kind of physical activity at least once a week. “Habit of smoking” was defined as currently smoking cigarettes or another tobacco product. ‘Habit of drinking alcohol’ was defined as more than 20 g/day alcohol intake. The differences of characteristics between the patients with and without usage of convenience stores were evaluated by Student’s t-test, Mann–Whitney U test or Chi-squared test separated by gender.

**Table 2 nutrients-11-01212-t002:** Dietary intake of study participants.

	Men			Women		
	Usage of Convenience Stores Three or More Times Per Week		*p*	Usage of Convenience Stores Three or More Times Per Week		*p*
*N*	(+) *n* = 24	(−) *n* = 182		(+) *n* = 9	(−) *n* = 152	
Total energy intake (kcal/kg IBW/day)	29.2 (21.6−38.1)	29.4 (25.0−35.3)	0.710	39.3 (27.7−42.9)	27.7 (22.8−33.5)	0.071
Protein intake (% Energy)	14.7 (12.5−16.3)	15.7 (13.7−17.8)	0.062	15.0 (13.7−16.5)	17.3 (15.7−20.2)	0.005
Fat intake (% Energy)	27.0 (23.6−28.5)	27.3 (24.1−31.0)	0.449	22.8 (19.9−30.3)	29.4 (26.6−34.0)	0.083
Carbohydrate intake (% Energy)	51.9 (46.6−56.8)	52.0 (44.8−57.3)	0.930	60.8 (51.5−64.5)	51.1 (45.4−55.8)	0.048
Dietary fiber intake (g/day)	9.9 (7.8−12.6)	11.7 (9.0−15.3)	0.102	9.1 (7.9−20.4)	11.0 (8.8−13.9)	0.987
Intake of carbohydrate to fiber ratio	21.5 (20.0−29.3)	19.9 (15.7−25.0)	0.052	21.0 (18.9−23.9)	16.2 (13.8−20.3)	0.017
Protein intake (g/kg IBW/day)	1.0 (0.9−1.3)	1.1 (0.9−1.4)	0.304	1.3 (1.1−1.5)	1.2 (1.0−1.6)	0.631
Fruit and vegetable intake (g/1000 kcal)	132 (102−191)	192 (128−267)	0.019	178 (132−207)	239 (172−313)	0.063
Pulses intake (g/1000 kcal)	23.6 (13.9−32.8)	27.6 (17.2−43.5)	0.093	18.8 (10.5−31.4)	38.8 (21.6−61.2)	0.011
Salt intake (g/day)	11.2 (8.8−13.4)	11.0 (8.9−13.0)	0.650	11.0 (7.5−13.5)	9.1 (7.4−11.6)	0.517
PRAL (mEq/day)	12.8 (2.5−18.9)	7.0 (0.1−14.2)	0.039	12.7 (5.2−16.8)	3.4 (−1.7−10.9)	0.122
NEAP (mEq/day)	55.2 (45.4−65.2)	48.9 (42.3−56.8)	0.013	56.9 (52.6−59.8)	46.3 (40.9−54.0)	0.050
Low-fat dairy products (g/1000 kcal)	0.0 (0.0−0.0)	0.0(0.0−58.2)	0.009	0.0 (0.0−0.0)	0.0 (0.0−58.0)	0.057
Processed meats (g/1000 kcal)	4.9 (1.5−8.0)	2.9 (1.4−6.5)	0.392	8.3 (2.8−10.6)	2.8 (1.6−7.2)	0.140
Confectionery (g/1000 kcal)	24.0 (11.5−37.6)	23.9 (10.0−41.3)	0.840	41.9 (27.1−48.1)	29.6 (13.9−45.4)	0.154

IBW, ideal body weight; PRAL, potential renal acid load score; NEAP, net endogenous acid production score. The differences of dietary intake between the patients with and without usage of convenience stores were evaluated by Mann–Whitney U test or Chi-squared test separated by gender.

**Table 3 nutrients-11-01212-t003:** Logistic regression analysis on the prevalence of hypertension.

	Odds Ratio (95% Confidence Interval)	*p*
Male	1.34 (0.74−2.42)	0.330
Age (years)	1.04 (1.01−1.07)	0.019
Duration of diabetes (years)	1.00 (0.97−1.03)	0.859
Body mass index (kg/m^2^)	1.19 (1.09−1.23)	<0.001
Hemoglobin A1c (mmol/mol)	0.99 (0.96−1.02)	0.467
Energy intake (kcal/kg ideal body weight/day)	1.01 (0.97−1.06)	0.565
Salt intake (g/day)	0.94 (0.84−1.07)	0.356
Habit of drinking alcohol	1.07 (0.44−2.61)	0.887
Habit of exercise	0.84 (0.48−1.45)	0.528
Habit of smoking	1.15 (0.49−2.72)	0.752
Usage of convenience stores three or more times per week	5.01 (1.12−22.50)	0.035

“Habit of exercise” was defined as performing any kind of physical activity at least once a week. “Habit of smoking” was defined as currently smoking cigarettes or another tobacco product. “Habit of drinking alcohol” was defined as more than 20 g/day alcohol intake.

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
