# Peer review of "Frequent Usage of Convenience Stores is Associated with Low Diet Quality"

_nutrients, 2019, doi:10.3390/nu11061212_

Round 1
Reviewer 1 Report
Type manuscript: Article – original research manuscript.
Title manuscript: Frequent Usage of Convenience Stores is Associated with Low Diet Quality
Short characteristic manuscript:
In the manuscript “Frequent Usage of Convenience Stores is Associated with Low Diet Quality” Authors tried determine the relationship between the usage of the convenience stores and dietary habits or the prevalence of hypertension among patients with type 2 diabetes. The Authors provided valuable information about, how often patients really buy meals from convenience stores instead of the number or density of convenience stores. The research have innovative character and deserve recognition.
General comments for Authors:
Generally the manuscript provides valuable information. However, I have some remarks.
First of all, the authors should not put following words in the abstract: background, methods, results and conclusions. Of course, these elements should be included in the abstract, but there is no need to put these words.
The authors excluded patients with very low or very high energy intake (<600 or="">4000 kcal/ day), but did not explain why? This is very important information and the authors should take it into account.
Did the authors carry out a test to determine if the distribution of variables is normal or does it deviate from normal? This information should be in the section Statistical analysis.
Statistical analysis is not described in detail, too little information is included in it. How were the significant variables that were related in logistic regression determined?
Comments and detailed suggestions for Authors:
Line 121-126;
This information should be in the Patients and Study Design section. In addition, much of this information is replicated with the information contained in the Patients and Study Design section. This part of the manuscript needs correction.
Table 3;
p-value should be written with a lowercase letter.
Author Response
Response to Reviewer 1
1. First of all, the authors should not put following words in the abstract: background, methods, results and conclusions. Of course, these elements should be included in the abstract, but there is no need to put these words.
Response
Thank you for your suggestion. According to your suggestion, we have revised Abstract.
2. The authors excluded patients with very low or very high energy intake (<600 or="">4000 kcal/ day), but did not explain why? This is very important information and the authors should take it into account.
Response
Thank you for your comment. We excluded the patients with very low or very high energy intake, because these calories intakes are unreliable. According to your comment, we have added this explanation in the Materials and Methods, described as below.
“Among the 438 patients (244 men and 194 women) who answered both the BDHQ and the questionnaire about convenience stores, we excluded the following 29 patients with the following characteristics: incomplete questionnaires (14 men and 7 women), extremely low or high energy intake (<600 or="">4000 kcal/day) [12] (3 men and 4 women) because these calorie intakes are unreliable.”
3. Did the authors carry out a test to determine if the distribution of variables is normal or does it deviate from normal? This information should be in the section Statistical analysis.
Response
Thank you for your comment. According to your comment, we have performed Shapiro-Wilk test to investigate the distribution of variables. We have changed Table 1 and Materials and Methods section described as below.
“To investigate the distribution of variables, we have performed Shapiro-Wilk test. Continuous variables were described as the mean (standard deviation) or median (interquartile range), and categorical variables were presented as the number. The differences of characteristics and dietary intake between the patients with and without usage of convenience stores were evaluated using Student's t-test, Mann–Whitney U test or chi-squared test.”
4. Statistical analysis is not described in detail, too little information is included in it. How were the significant variables that were related in logistic regression determined?
Response
Thank you for your comment. According to your comment, we have added the information of how we choose the variables, described as below.
“The following factors were considered as independent variables: age, sex, body mass index (BMI) [25], duration of diabetes, hemoglobin A1c [26], energy intake, salt intake [27], habit of drinking alcohol [28], habit of exercise [29], and habit of smoking [30].”
25. He YH, Jiang GX, Yang Y, Huang HE, Li R, Li XY, Ning G, Cheng Q. Obesity and its associations with hypertension and type 2 diabetes among Chinese adults age 40 years and over. Nutrition. 2009;25:1143–9.
26. Ha SK. Dietary salt intake and hypertension. Electrolyte Blood Press. 2014;12:7–18.
27. Fuchs FD, Chambless LE, Whelton PK, Nieto FJ, Heiss G. Alcohol consumption and the incidence of hypertension: The Atherosclerosis Risk in Communities Study. Hypertension. 2001;37:1242–50.
28. Cornelissen VA, Fagard RH. Effects of endurance training on blood pressure, blood pressure-regulating mechanisms, and cardiovascular risk factors. Hypertension. 2005;46:667–75.
29. Ezzati M, Henley SJ, Thun MJ, Lopez AD. Role of smoking in global and regional cardiovascular mortality. Circulation. 2005;112:489–97.
5. Line 121-126; This information should be in the Patients and Study Design section. In addition, much of this information is replicated with the information contained in the Patients and Study Design section. This part of the manuscript needs correction.
Response
Thank you for your suggestion. According to your suggestion, we have changed the Patients and Study Design section, described as below.
“Among the 438 patients (244 men and 194 women) who answered both the BDHQ and the questionnaire about convenience stores, we excluded the following 29 patients with the following characteristics: incomplete questionnaires (14 men and 7 women), extremely low or high energy intake (<600 or="">4000 kcal/day) [12] (3 men and 4 women) because these calorie intakes are unreliable, no data in biochemical tests (one man), and no type 2 diabetes (20 men and 22 women). Thus, the final study population was 367 patients (206 men and 161 women).”
6. Table 3; p-value should be written with a lowercase letter.
Response
Thank you for your suggestion. According to your suggestion, we have described p-value as a lowercase letter.
Reviewer 2 Report
Thank you for the opportunity to review this manuscript which describes findings from a study considering associations between convenience store usage with diet and lifestyle behaviors among Japanese individuals with type 2 diabetes. An important strength of the study is the consideration of average convenience store usage during the week, moving beyond previous literature which only considers presence or density of convenience stores in neighborhoods but not actual usage. In addition, individuals with type 2 diabetes are an important group to focus on whose convenience store usage has not been considered previously. I have provided some overall comments of areas for clarification or additional information that I think would help to strengthen the manuscript, as well as some specific line comments.
Overall
· Overall, the manuscript could benefit from additional editing for grammar/English
· The authors note the median age of the sample is 68, suggesting this is a population of older adults. It would be useful to provide some additional context as to how convenience store usage may function for older adults, and what the significant age differences the authors found might mean.
· Although the authors note in the Introduction and Discussion that convenience stores are more common in Japan than in other counties, they found that only 33 individuals in their sample used convenience stores 3 or more times per week, suggesting usage is not that high at least in their sample. How does this compare to other countries? Is it the case that convenience stores are more common, but Japanese individuals do not use them any more frequently? Related, the analyses considered separately for gender lead to very small groups of the convenience store users for women (n=9). It would be useful to at least note this in the limitations, as results from only 9 individuals are likely not generalizable.
Abstract
· Lines 11-12: The first sentence of the abstract seems to state a previous research finding but does not have a reference, is this in previous research studies or the current findings?
Introduction:
· Lines 51-52: Can you provide more background on why it is specifically important to consider convenience store usage among individuals with type 2 diabetes?
Method
· Lines 90-99: During what time points in the study were the blood, urine, and blood pressure measurements collected? How do these time points relate to the self-reports of dietary intake and behavior for the previous month?
Discussion
· The authors noted they considered convenience store usage dichotomously as three or more times per week or less than three times per week based on previous research findings. While this seems to be a valid way to consider this variable, I am curious if they could provide additional consideration and discussion of what the “three times per week” variable means conceptually for a Japanese population. Would this include meals or snacks? How does this fit into a regular diet?
Author Response
Response to reviewer 2
1. Overall, the manuscript could benefit from additional editing for grammar/English
Response
According to your suggestion, our manuscript was checked by a native English speaking colleague.
2. The authors note the median age of the sample is 68, suggesting this is a population of older adults. It would be useful to provide some additional context as to how convenience store usage may function for older adults, and what the significant age differences the authors found might mean.
Response
Thank you for your valuable comment. As you say, the median age of the people who used convenience stores 3 or more times per week is 68 and this median age is older than that of the people who did not use. The possible reason why the median age of the people who use convenience stores 3 or more times per week is older might be that the older patients with type 2 diabetes is difficult to cook themselves. According to your comment, we have added this point in the Discussion section, described as below.
“The median age of the people who used convenience stores 3 or more times per week is 68 and this median age is older than that of the people who did not use. The possible reason why the median age of the people who use convenience stores 3 or more times per week is older might be that the older patients with type 2 diabetes is difficult to cook themselves.”
3. Although the authors note in the Introduction and Discussion that convenience stores are more common in Japan than in other counties, they found that only 33 individuals in their sample used convenience stores 3 or more times per week, suggesting usage is not that high at least in their sample. How does this compare to other countries? Is it the case that convenience stores are more common, but Japanese individuals do not use them any more frequently? Related, the analyses considered separately for gender lead to very small groups of the convenience store users for women (n=9). It would be useful to at least note this in the limitations, as results from only 9 individuals are likely not generalizable.
Response
Thank you for your comment and suggestion. As you say, the case of frequent usage of convenience is not so high and to compare the proportion of frequent usage of convenience of Japanese with that of other countries is important. Unfortunately, however, no previous studies exanimated the frequent usage of convenience. Thus, we cannot compare the proportion of frequent usage of convenience of Japanese with that of other countries. We found that only 33 individuals in our sample used convenience stores 3 or more times per week. We do not think that the number is not high, because usage of convenience stores 3 or more times per week seems to be a quite high frequency.
In addition, as you say, the number of women of convenience store users is small. We have mentioned these points as one of the limitations of this study in the Discussion section, described as below.
“Fourth, to compare the proportion of frequent usage of convenience of Japanese with that of other countries is important. Unfortunately, however, no previous studies exanimated the frequent usage of convenience. In addition, the number of women of convenience store users is small. Thus, results from only 9 individuals are likely not generalizable.”
4. Abstract: Lines 11-12: The first sentence of the abstract seems to state a previous research finding but does not have a reference, is this in previous research studies or the current findings?
Response
Thank you for your comment. The first sentence of the abstract is the results of previous research studies. Thus, we have changed the sentence as below.
“Previous studies revealed that density of convenience stores in the neighborhood was associated with chronic diseases.”
5. Introduction: Lines 51-52: Can you provide more background on why it is specifically important to consider convenience store usage among individuals with type 2 diabetes?
Response
Thank you for your suggestion. According to your suggestion, we have added the sentence described as below.
“In addition, diet is an essential treatment target for patients with type 2 diabetes [9]. Thus, to investigate the convenience store usage in patients with type 2 diabetes has an important meaning.”
9. Evert AB, Boucher JL, Cypress M, Dunbar SA, Franz MJ, Mayer-Davis EJ, Neumiller JJ, Nwankwo R, Verdi CL, Urbanski P, Yancy WS Jr; American Diabetes Association. Nutrition therapy recommendations for the management of adults with diabetes. Diabetes Care. 2013;36:3821-42.
6. Method: Lines 90-99: During what time points in the study were the blood, urine, and blood pressure measurements collected? How do these time points relate to the self-reports of dietary intake and behavior for the previous month?
Response
Thank you for your comment. We collected the data of blood, urine and blood pressure when the self-reports of dietary intake were performed. Thus, the effect of the past month’s behavior might be appeared in the data of blood, urine and blood pressure, we think. According to your comment, we have added this point in the Materials and Methods section, described as below.
“Blood samples and urine samples were collected in the morning for biochemical measurements, when the self-reports of dietary intake were performed.”
“Blood pressure measurement was performed automatically using a device (HEM-906; OMRON, Kyoto, Japan) in a quiet space after 5 minutes of rest, when the self-reports of dietary intake were performed.”
7. Discussion: The authors noted they considered convenience store usage dichotomously as three or more times per week or less than three times per week based on previous research findings. While this seems to be a valid way to consider this variable, I am curious if they could provide additional consideration and discussion of what the “three times per week” variable means conceptually for a Japanese population. Would this include meals or snacks? How does this fit into a regular diet?
Response
Thank you for your valuable comment. In this study, we asked the patients with type 2 diabetes about “Did you buy meals from convenience stores three or more times per week?” Thus, this might be indicated that the patients who answered yes eat meals of convenience store three or more times per week and eat meals of convenience store three or more times per week was associated with low quality of diet and the prevalence of hypertension.